# Integrin Targeting Enhances the Antimelanoma Effect of Annexin V in Mice

**DOI:** 10.3390/ijms24043859

**Published:** 2023-02-15

**Authors:** Jingyi Zhu, Xiangning Li, Wenling Gao, Jian Jing

**Affiliations:** 1Beijing Key Lab of Biotechnology and Genetic Engineering, College of Life Sciences, Beijing Normal University, Beijing 100875, China; 2College of Life Sciences, Beijing Normal University, Beijing 100875, China

**Keywords:** malignant melanoma, annexin V, lebestatin, antitumor

## Abstract

Malignant melanoma, an increasingly common form of skin cancer, is a major threat to public health, especially when the disease progresses past skin lesions to the stage of advanced metastasis. Targeted drug development is an effective strategy for the treatment of malignant melanoma. In this work, a new antimelanoma tumor peptide, the lebestatin–annexin V (designated LbtA5) fusion protein, was developed and synthesized by recombinant DNA techniques. As a control, annexin V (designated ANV) was also synthesized by the same method. The fusion protein combines annexin V, which specifically recognizes and binds phosphatidylserine, with the disintegrin lebestatin (lbt), a polypeptide that specifically recognizes and binds integrin α1β1. LbtA5 was successfully prepared with good stability and high purity while retaining the dual biological activity of ANV and lbt. MTT assays demonstrated that both ANV and LbtA5 could reduce the viability of melanoma B16F10 cells, but the activity of the fusion protein LbtA5 was superior to that of ANV. The tumor volume growth was slowed in a mouse xenograft model treated with ANV and LbtA5, and the inhibitory effect of high concentrations of LbtA5 was significantly better than that of the same dose of ANV and was comparable to that of DTIC, a drug used clinically for melanoma treatment. The hematoxylin and eosin (H&E) staining test showed that ANV and LbtA5 had antitumor effects, but LbtA5 showed a stronger ability to induce melanoma necrosis in mice. Immunohistochemical experiments further showed that ANV and LbtA5 may inhibit tumor growth by inhibiting angiogenesis in tumor tissue. Fluorescence labeling experiments showed that the fusion of ANV with lbt enhanced the targeting of LbtA5 to mouse melanoma tumor tissue, and the amount of target protein in tumor tissue was significantly increased. In conclusion, effective coupling of the integrin α1β1-specific recognition molecule lbt confers stronger biological antimelanoma effects of ANV, which may be achieved by the dual effects of effective inhibition of B16F10 melanoma cell viability and inhibition of tumor tissue angiogenesis. The present study describes a new potential strategy for the application of the promising recombinant fusion protein LbtA5 in the treatment of various cancers, including malignant melanoma.

## 1. Introduction

Annexin V, which belongs to the annexin superfamily, was first discovered as an anticoagulant protein and soon attracted attention as a novel biomarker for apoptosis due to its high binding affinity for phosphatidylserine (PS) (dissociation constant Kd = 15 nM) in the presence of calcium. PS is restricted to the inner leaflet of the plasma membrane in healthy cells. When cells become apoptotic, PS is flipped to the outer leaflet, which is a major ‘eat-me’ signal for phagocytes, such as macrophages. However, in tumor cells, high levels of PS are present on the outer leaflet of the cell membrane, and PS externalization is not related to apoptosis [1]. PS is present on the luminal surface of vascular endothelial cells in various tumors but not in normal tissues [2]. The high affinity of ANV for PS and the high expression of externalized PS by tumor cells allow ANV to be used as a tool for targeting tumor tissue.

Angiogenesis is the basis for tumor growth, invasion, and metastasis. Solid tumors usually have a strong ability to induce new blood vessel formation, which provides more nutrients and oxygen to nourish the tumor tissue and help it metastasize. Angiogenesis in healthy tissue is tightly controlled by the balance between pro- and antiangiogenic factors, but, in tumor tissue, this balance is usually disrupted, and the angiogenic switch is almost always activated, resulting in tumor tissue with a strong angiogenic phenotype. In 1975, Folkman proposed that inhibition of tumor angiogenesis would be a new strategy for the treatment of cancer [3].

Induction of angiogenesis is key to the development of most human tumors, including melanoma. Numerous studies have shown that human melanoma induces angiogenesis [4]. High expression of several angiogenic factors, including vascular endothelial growth factor, basic fibroblast growth factor, and interleukin-8, has been detected in primary cutaneous melanoma, and the importance of these factors in promoting melanoma angiogenesis and metastasis has been demonstrated in tumor xenograft models [5]. Based on these findings, a number of angiogenesis inhibitors targeting human melanoma are under investigation [6].

Integrins are essential mediators and regulators of physiological and pathological tumorigenesis and angiogenesis. Integrins exist as an α:β heterodimeric complex of transmembrane proteins, one of the adhesion receptor superfamilies, and play vital roles in regulating cell–matrix and cell–cell interactions [7]. Integrins play a key role in tumor development by supporting cancer cell proliferation, angiogenesis, and metastasis.

Disintegrins are small cysteine-rich proteins found in a variety of snake venoms. These proteins selectively regulate integrin function and heterodimeric receptors involved in cell–cell and cell–matrix interactions, and have been extensively studied as therapeutic targets. Lebestatin (lbt), a kind of low-molecular-weight protein, was isolated from Tunisian snake (*Macrovipera lebetina*) venom. This protein has only 41 amino acids and is considered the shortest disintegrin yet described. Lbt contains a lysine-threonine-serine (KTS) motif in the integrin recognition loop and is a selective inhibitor of the binding of integrin α_1_β_1_ to type IV and type I collagens. Integrin α_1_β_1_, a receptor of laminin-1 and collagens, has been reported to be involved in cell migration and tumor angiogenesis [8]. Snake disintegrins were demonstrated to have potent antiangiogenic activity [9,10].

In this study, we used genetic engineering to develop a new fusion protein that links the C-terminus of the lebestatin disintegrin to the N-terminus of ANV via a specific linker. Due to the high affinity of ANV for PS and the high expression of ectopic PS in tumor tissues, the disintegrin lbt, which can inhibit angiogenesis, accumulates in tumor tissue. Thus, we postulate that the fusion protein LbtA5 shows more promise than ANV for the treatment of melanoma.

## 2. Results

### 2.1. Preparation and Determination of Recombinant LbtA5 Fusion Protein

We used the recombinant expression vectors of ANV previously constructed in our laboratory for the expression of GST-ANV, and we constructed the LbtA5 recombinant expression vector on this basis. The protein was then purified using affinity chromatography. The results of SDS–PAGE electrophoresis showed that most of the target protein was present in the supernatant after ultrasonic centrifugation for both GST-ANV and GST-LbtA5, and the amount of target protein in the precipitate was low (Figure 1). The theoretical molecular weight of GST-LbtA5 was 66.3 kDa, and the electrophoretic results showed that there was a clear band at 70 kDa after purification, which was consistent with the theoretical value. The theoretical molecular weight of GST-ANV was 61.9 kDa, and the electrophoretic results showed that the protein aggregated at approximately 55 kDa after purification, which was consistent with the theoretical value. Moreover, the purity of GST-ANV and GST-LbtA5 was above 95% after GST affinity chromatography.

Next, the target proteins were cleaved by using PreScission Protease (PSP) prepared in the laboratory. The results showed that GST-LbtA5 and GST-ANV were completely cleaved after 1 h of PSP digestion, with theoretical values of 40.3 kDa for LbtA5, 35.9 kDa for ANV, and 26 kDa for GST. The results showed that the purity of the target protein was above 98% and could be used for subsequent experiments. Our studies have shown that, combined with GST column affinity chromatography technology, ANV and LbtA5 recombinant proteins were obtained with relatively good purification, and the preparation yield was also relatively high.

### 2.2. LbtA5 Retains Phospholipid Binding Properties

The main characteristic of ANV is its high affinity to phospholipids; thus, ANV plays an important role in membrane repair, anticoagulation, and apoptosis, so we examined the ability of ANV or LbtA5 to bind to phosphatidylserine (PS) or phosphatidylcholine (PC). As shown in Figure 2a, the fusion protein LbtA5 exhibited the same properties as ANV, binding specifically to PS but not PC. With increasing concentrations of LbtA5 and ANV, the absorbance values of the PS-encapsulated pores increased at OD_450_, showing a dose-dependent effect, while the absorbance values of the PC group remained essentially unchanged. In addition, the binding of ANV and LbtA5 to PS and Ca^2+^ exhibited dose-dependent effects and saturation at a PS concentration of 2 μg and a Ca^2+^ concentration of 3 mM, respectively (Figure 2b,c). Our experimental results illustrated that the fusion protein LbtA5 did not block the binding site of ANV to PS. This finding suggested that the fusion protein LbtA5 was able to function properly as ANV and was an important basis for subsequent experiments.

### 2.3. LbtA5 Retained the Integrin Specificity

First, we obtained purified GST-lebestatin (Figure 3a). Previous studies have shown that the disintegrin lbt specifically binds to the α_1_β_1_ integrin, thereby inhibiting the binding of other ligands to α_1_β_1_. α_1_β_1_ binds specifically to Col-IV, and α_1_β_1_ is highly expressed in PC12 cells. HT29 cells were used as a control, and HT29 cells expressed the integrins α_v_β_5_, α_v_β_6_, and α_6_β_4_. Among them, α_6_β_4_ binds specifically to laminin (LN) and promotes cell adhesion. In the HT29 group, the number of cells decreased after the addition of different concentrations of lbt, LbtA5, and ANV, but there was no significant overall change (Figure 3c,e). This finding indicates that these three proteins did not inhibit the binding of HT29 cells to the ligand LN and that lbt is not a specific antagonist of the integrins expressed by HT29 cells. Compared with that of the control group, the number of PC12 cells in the 96-well plates after Col-IV treatment was significantly increased, indicating that the integrins expressed by PC12 cells were able to bind specifically to Col-IV. There was no significant change in the number of cells after PC12 cells were co-incubated with different concentrations of ANV, which remained consistent with the results of the HT29 group. However, after addition of lbt and LbtA5 and co-incubation with PC12, the absorbance value decreased, indicating that the number of PC12 cells decreased (Figure 3b,d). The increase in protein concentration showed a dose-dependent effect. The results indicate that lbt and LbtA5 can specifically bind integrin α_1_β_1_ expressed by PC12 cells. In conclusion, we demonstrated that LbtA5 maintained the disintegrin activity of lbt.

### 2.4. LbtA5 Inhibits the Proliferation and Migration of Melanoma B16F10 Cells

B16F10 cells were treated with LbtA5 and ANV, three protein concentrations (0.1 nM, 1 nM and 10 nM) were used, and cell viability was measured by the MTT method after 24 and 48 h. Figure 4a shows that the cell viability of the ANV-treated group increased after 24 h, and the B16F10 cell viability was proportional to the ANV concentration. However, the cell viability decreased after 48 h. Compared to that of the LbtA5-treated group, B16F10 cell viability decreased after both 24 and 48 h. Among them, there was a significant difference (*p* < 0.05) compared to the control group after 48 h, and the treatment showed a dose-dependent effect. The experimental results indicate that a higher concentration and longer duration of action of LbtA5 had a greater effect (inhibition) on the viability of B16F10 cells. In addition, the inhibitory effect of LbtA5 on B16F10 cell viability was superior to that of ANV at the same protein concentration and duration of action.

To assess the effect of LbtA5 and ANV on B16F10 cell migration, we used a cell scratch assay, treating B16F10 cells with different concentrations of the two proteins for 24 h and 48 h, and the scratch healing results are shown in Figure 4b. After 24 h, the scratch was narrower in the cells without protein treatment, and scratch healing became worse as the protein concentration increased in the experimental group. The scratched area was quantified using ImageJ software, and the results showed that the cell migration rate of the control group was 50%, which did not change significantly when the protein concentration was 0.1 nM and decreased in a dose-dependent manner when the concentration reached 1 nM and 10 nM. When the protein concentration was 1 nM, LbtA5 inhibited cell migration significantly more than ANV. The cell scratching assay fully illustrated that both ANV and LbtA5 had a very obvious inhibitory effect on melanoma cell migration, and the cell migration rate after treatment with a certain concentration of LbtA5 was significantly lower than that after treatment with ANV.

We further verified the effect of LbtA5 and ANV on apoptosis of B16F10 cells using the annexin V-FITC/PI double-staining method. The results are shown in Figure 4. The apoptosis rate of the cells did not change significantly after 24 h of adding LbtA5 and ANV (Figure 4c). The above data indicate that although LbtA5 could not affect the apoptosis of B16F10 cells, it could inhibit cell proliferation and migration, and, in particular, the inhibitory effect of LbtA5 was stronger than that of ANV.

### 2.5. LbtA5 Inhibits Tumor Growth in a Mouse Xenograft Model Using B16F10 Cells

A mouse xenograft model with murine melanoma B16F10 cells was established. These six treatment groups were assigned to receive vehicle (e.g., PBS) as a negative control, the anticancer drug DTIC as a positive control, and two concentrations of annexin V and two dosages of LbtA5 for the treatments. In the B16F10 model, compared to those in the PBS group, the mice in the high-concentration LbtA5 and DTIC groups showed a flat growth trend, although their body weight increased slightly on the thirteenth day of treatment (DTIC 9.58%; LbtA5 16.30%; PBS 28.55%). This finding suggests that these treatments may have had a relatively strong effect on the mouse organism, resulting in slow body weight gain in mice. After 13 days of treatment, Figure 5b shows that LbtA5 (139 μmol/kg) with the same concentration inhibited the tumor growth rate better than ANV. There was no significant difference in melanoma inhibition with high concentrations of LbtA5 (278 μmol/kg) compared to the positive control DTIC.

### 2.6. LbtA5 Induced Tumor Cell Necrosis In Vivo

The mice were dissected, tumor tissues were collected, and sections were made for H&E staining. As shown in Figure 5c, the tumor cells in the PBS group were densely arranged with clear outlines and less cytoplasm. The tumor cells in the experimental groups all had different degrees of pathological necrosis. The tumor cells in the low-concentration ANV group were widely spaced, and the necrotic area was accompanied by bleeding, while the LbtA5 group had some neutrophil aggregates, while loose cells with poor contours could be observed. Tumor tissue necrosis was more obvious in the high-concentration ANV and LbtA5 groups, and the cells were scattered and not in pieces, which might be related to collagen fiber lysis. There were many aggregated neutrophils in the LbtA5 group, the degree of cell fragmentation was higher, and red-stained cell debris appeared. Histomorphological observation showed that both LbtA5 and ANV had certain effects on tumor growth, and the higher the concentration, the more serious the tumor necrosis; at the same injection amount, the destructive effect of LbtA5 on tumor tissue was stronger than that of ANV.

### 2.7. LbtA5 SPECIFICALLY Inhibits Angiogenesis in Mouse Melanoma Tumors

To verify whether ANV and LbtA5 affect tumor angiogenesis, we assessed the vascular density of B16F10 tumors by immunofluorescence using a rabbit polyclonal antibody against desmin (a tumor vascular marker). Red linear fibrous vascular tissue was visible in tumor tissues of the PBS group (Figure 6a), and vascular density was significantly reduced in the low-concentration ANV group, while no vascular tissue was visible in the tumors treated with high-concentration ANV and LbtA5. These results indicate that LbtA5 and ANV may hinder tumor growth by inhibiting angiogenesis, indicating that LbtA5 fused with lbt shows more significant inhibition of tumor angiogenesis in vivo than ANV. In addition, we examined VEGF expression in B16-F10 cells by western blotting (Figure 6b,c). Interestingly, the expression of VEGF did not decrease significantly with increasing concentrations of the target protein. Therefore, the experimental results suggest that LbtA5 and ANV do not inhibit melanoma angiogenesis and thus tumor growth by decreasing the level of VEGF expression in cultured B16-F10 cells in vitro.

### 2.8. Targetting of LbtA5 Mice Melanoma Tumor Tissue

To investigate the location of LbtA5 and ANV in the mouse organism, we injected FITC-labeled ANV(fANV) and LbtA5(fLbtA5) into mice carrying B16F10 solid tumors via tail vein, respectively, and observed the localization of fluorescein-labeled targeting proteins in mouse tumors under a fluorescence microscope after 30 min. As shown in Figure 7a–c, we successfully obtained FITC fluorescently labeled fANV and fLbtA5 proteins. Compared with the control group, the green fluorescence density and intensity in the tumor tissues gradually increased with the increase in the injected LbtA5 amounts with ANV as control. It could be observed that, when the injected amounts reached 417 μmol/kg and 834 μmol/kg, for the same injected amounts, the green fluorescence density in the LbtA5 group was stronger than that in the ANV group (Figure 7d). The results suggest that LbtA5 and ANV can both act directly on mice melanoma, but, obviously, LbtA5 exhibited stronger melanoma tumor targeting, which makes the target protein converge more in the melanoma tissue.

## 3. Discussion

Melanoma is a type of tumor that originates from melanocytes and has a high degree of malignancy. In recent years, the incidence of melanoma has been increasing, affecting younger patients over time [11]. Because traditional chemotherapy drugs cannot distinguish cancer cells from normal cells, it is difficult to administer sufficiently high doses to prevent relapse and metastasis after treatment without significant systemic toxicity [12]. For this reason, researchers worldwide have developed natural product drugs for targeted therapeutic strategies to address melanoma [13]. Tumor-targeted therapies have gradually advanced through preclinical and clinical testing, including targeted delivery carriers and targeted drugs [14]. In comparison with monoclonal antibodies, peptides that target tumors have the characteristics of high tissue-penetrating power, high selectivity for the tumor cell surface, and simple synthesis, which are expected to improve the treatment of cancers. At present, the RGD peptide shows promise as a targeting molecule, especially if considered in the context of integrin α_V_β_3_, which is highly expressed on the surface of tumor vascular endothelial cells and has been shown to inhibit tumor growth and initiate the tumor apoptotic process [15,16]. Integrin α_1_β_1_ plays a unique role in vascular proliferation, providing support for VEGF signaling, endothelial cell migration and tumor angiogenesis [17]. Among these, β_1_ integrin mediates multiple cellular functions and is a candidate target for interaction with the extracellular matrix (ECM), playing an important antitumor role. Inhibition of β_1_ resulted in a significant reduction in breast cancer cells [18], and downregulation of β_1_ inhibited metastasis in hepatocellular carcinoma cells [19]. Inhibition of integrin α_1_β_1_ reduced tumor cell invasion in the ECM in chemotherapy-treated mice. Treatment with obtustatin, a KTS disintegrin, a specific inhibitor of α_1_β_1_, reduced the movement of triple-negative breast cancer cells inoculated on a V-type collagen matrix [20] and reduced the migration and invasion of neuroblastoma cells with high TRPM2 expression [21], indicating the potential of α_1_β_1_ to increase cancer cell metastasis. These findings suggest that integrin α_1_β_1_ is an important target for intervention in cancer therapy. 

We designed a new type of antimelanoma tumor agent, LbtA5, through the ligation of ANV with the specific α1β1 integrin receptor antagonist lebestatin. ANV is a calcium-dependent phospholipid-binding protein, and ANV binds with high affinity to PS on the surface of apoptotic and necrotic cells, thereby impairing its uptake by macrophages. Anionic phospholipids, principally PS, are specifically exposed on tumor endothelial cells, possibly in response to oxidative stresses present in the tumor microenvironment [22,23]. Although enhanced PS externalization in cancer is widely accepted, the exact mechanism remains unclear [24]. Tumor cells and tumor vasculature alter their total phospholipid concentrations, have more PS in the plasma membrane, and have up to sevenfold more externalized PS than healthy cells [1]. Thus, the increased externalization of PS in many cancer types makes it a potential biomarker for cancer therapy [25,26,27]. Monoclonal antibodies that block PS interactions with its receptors have demonstrated antitumor activity in mouse tumor models [28]. Our studies show that the ANV portion of the fusion protein LbtA5 still binds specifically to PS. 

ANV is a calcium-dependent phospholipid-binding protein, and ANV binds with high affinity to PS on the surface of apoptotic and necrotic cells, thereby impairing their uptake by macrophages. Currently, many studies on ANV fusion proteins have been carried out. ANV was fused with the AH5 peptide sequence (SPSYAYHQF), a reported tumor antigenic epitope of CT26, to generate the ANV-AH5 fusion protein. ANV-AH5 administration following cisplatin resulted in the best tumor control, prolonged survival of mice, and the generation of the strongest systemic and tumor-infiltrating AH5-specific CD8+ T-cell response compared to those with other treatment regimens [29]. The fusion protein mCTH-ANV, developed by Roger G. Harrison et al., has no therapeutic effect on tumors when used alone [30]. However, mCTH-ANV played a key role in an enzyme prodrug system developed by these researchers. The system consists of mCTH-ANV and SeMet (a nontoxic substrate). mCTH-ANV refers to mutant cystathionine gamma-lyase fused to ANV, thereby targeting PS exposed to the tumor vascular system [31]. mCTH converts SeMet to toxic methyl selenol and generates reactive oxygen species, leading to tumor cell death. The results of the study showed that the enzyme prodrug system in combination with immunostimulants (anti-CD73 and anti-OX40) resulted in a 100% increase in survival 12 to 24 days after treatment, indicating that the enzyme prodrug system containing mCTH-ANXA5 acts synergistically with immunostimulants [31]. Annexin V-TRAIL (TP8), a fusion protein composed of ANV and tumor-necrosis-factor-related apoptosis-inducing ligand (TRAIL), effectively inhibited tumor growth in mouse xenografts from multiple cancer cell types, including non-small-cell lung cancer (A549), colon cancer (Colo205), and liver cancer (Bel7402) [32]. TRAIL, a member of the tumor necrosis factor (TNF) family, promotes selective tumor apoptosis by binding to death receptor (DR) 4 or 5 [33]. However, a large number of cancer cells are resistant to TRAIL-induced apoptosis, which is a major challenge for the treatment of tumors with TRAIL. In vitro experiments showed that the binding ability of TP8 to death receptors was not enhanced, and TP8 did not alter the expression levels of DR4 and DR5 [32]. A strategy relying on TP8 to enhance TRAIL resistance properties appears to be feasible, but the study did not clarify whether TP8 targets tumor tissues. Another study showed that the fusion protein ANV-IL2 was more effective in enhancing T-cell activation than recombinant IL-2 because interleukin-2 (IL-2) carrying ANV could deliver IL-2 to the surface of CD8 T cells where transient rearrangement of phospholipids occurred [34]. This finding suggests that the ANV fusion protein is an effective way to target PS, and, with the help of the high affinity of ANV for PS, ANV could play a role in tumor therapy as well as in the immune response. It has been suggested that tumor-derived microvesicles favor the establishment of melanoma metastasis in a PS-dependent manner [35], and ANV treatment inhibited melanoma size in mice [2]. Our study demonstrates that our fusion protein LbtA5 has the ability to inhibit melanoma in mice more effectively than ANV. 

We demonstrated by a cell adhesion assay that LbtA5 maintained the cell adhesion inhibitory activity of lbt. Disintegrins competitively bind to disintegrin receptors and block the interaction of other ligands. This class of integrins is very important in cell adhesion. The attachment of metastatic cells is highly dependent on successful induction by transformed cells of normal cell–cell and cell–matrix interactions so that newly established colonies can integrate with circulatory and other physiological systems. These interactions are extremely complex and involve surface interactions between tumor cells and surrounding tissues [36,37]. Expression analysis at different stages of melanoma progression indicates that the levels of β_3_ and β_1_ integrins promote melanoma transition from the radial growth phase to the vertical growth phase [5]. KTS disintegrins are known to be integrin-β_1_-specific antagonists. At the cellular level, LbtA5 inhibited the proliferation and migration of B16F10 cells more effectively than ANV. Our study shows that LbtA5, a fusion protein that efficiently and specifically recognizes and binds integrin α_1_β_1_, is highly likely to act directly on such integrin receptors on B16F10 cell membranes, exerting its superior tumor-cell-suppressive activity over ANV alone.

However, our study also showed that LbtA5 inhibited tumor angiogenesis better than ANV. The development of a rich vascular network in the tumor microenvironment is crucial in melanoma progression, especially in the vertical growth phase. Lbt exhibited antiangiogenic effects in an 8-day-old chick embryo chorioallantoic membrane model [10]. Compared with ANV, LbtA5 was more effectively enriched in the vascular region of mouse melanoma, suggesting that LbtA5, which exhibited high α1β1 binding capacity, was more capable of inhibiting tumor growth by suppressing angiogenesis. Furthermore, the fluorescent tissue localization results showed that LbtA5 had better tumor targeting than ANV. Therefore, based on the above experimental results, showing that fused LbtA5 has a better effect of inhibiting melanoma in mice in vivo, this effect may be achieved by both direct targeting of tumor cells and inhibition of angiogenesis.

ANV was able to inhibit melanoma development in mice, but coupling the integrin-targeted protein structure significantly enhanced the biological antimelanoma effect of the fusion protein. Both the inhibitory effect on melanoma B16F10 cells and the content of the fusion protein in melanoma tumor tissues were significantly higher than those of ANV, indicating that the fusion protein LbtA5 has stronger melanoma tumor targeting. LbtA5 produced a stronger inhibitory effect on angiogenesis, which is also a very interesting finding, indicating that the targeting of integrin receptors can more effectively enhance the inhibition of tumor tissue angiogenesis. Angiogenesis is a complex physiopathological process involving numerous functional protein molecules. Interestingly, we found that neither the LbtA5 fusion protein nor the ANV monomer protein altered the VEGF content of mouse melanoma tumor tissues, suggesting that the inhibition of angiogenesis by LbtA5 is achieved through other pathways. However, it is indisputable that the biological antimelanoma effect of ANV was significantly improved after integrin receptor targeting, which will help the development of novel antimelanoma drugs in the future and provide a new idea and reference for pharmaceutical research and development in this area.

## 4. Materials and Methods

### 4.1. Protein Production and Purification

For generation of lebestatin, lbt was first amplified by PCR using two pairs of primers as templates:

Lbt1F:

5′-CGTCAGTGTAAACTGAAACCGGCTGGTACCACCTGTTGGAAAACCTCC-3′

Lbt1R:

5′-CAGTCACAGGATTTACCGGTAGAGTAGTGGGAGGTACGGGAGGTTTTCCA-3′Lbt2F: 5′-CCGCCATATGTGTACCACCGGTCCGTGTTGTCGTCAGTGTA-3′

Lbt2R: 5′-CGCGGGATCCTCAACCCGGGTAGGACGGACAGTCACAGGA-3′

The PCR conditions were set as follows: predenaturation at 94 °C for 5 min, denaturation at 94 °C for 45 s, annealing at 56 °C for 45 s, extension at 72 °C for 90 s (30 cycles), and final extension at 72 °C for 10 min. The reaction product was stored at 4 °C. The PCR product was lebestatin. Next, four Glys were attached to the C-terminus of lbt by PCR with the following primers:

LbtG-F: 5′-CGATGGATCCATGTGTACCACCGGTCCG-3′ (containing restriction sites for BamHI)

LbtG-R: 5′-ACCACCACCACCACCCGGGTAGGACGG-3′

The PCR product was LbtG. pGEX-6P-1-ANV was then used as a template to attach four Glys to the N-terminal end of ANV with the following primers. The plasmid pGEX-6p-1-annexin V was previously constructed in this lab.

GANV-F: 5′-GGTGGTGGTGGTGCACAGGTTCTCAGAGGC-3′

GANV-R: 5′-GCATCTCGAGCTATTAGTCGTCCTCTCC-3′

The PCR product was GANV. Finally, PCR was performed using LbtG and GANV as templates and LbtG-F and GANV-R as primers to obtain the lebestatin–annexin V gene fragment. The PCR product was cloned into the BamHI and XhoI sites of the pGEX-6P-1 vector. The PCR conditions were set as follows: predenaturation at 94 °C for 5 min, denaturation at 94 °C for 1 min, annealing at 56 °C for 1 min, extension at 72 °C for 2 min (30 cycles), and final extension at 72 °C for 10 min. The reaction product was stored at 4 °C. The DNA plasmids were confirmed by sequencing and transformed into *Escherichia coli* (BL21(DE3)). The GST fusion protein was induced and isolated as described previously. Protein concentrations were determined using BCA Protein Assay Reagent (Beyotime, Shanghai, China). The secondary structure of the target protein was detected using a circular dichroism spectrometer.

### 4.2. Phospholipid-Binding Assay of ANV and LbtA5

Ten milligrams of PS (Beyotime, Shanghai, China) or PC (Liweining, Beijing, China) was dissolved in 10 mL of chloroform, added to each microtiter well (2.5 g/well), and adsorbed on the polystyrene surface by evaporating off the solvent at 37 °C. The plate was blocked with 5% skim milk in Tris-buffered saline (TBS) for 2 h at 37 °C. Protein samples such as LbtA5 fusion protein at various concentrations were added to the wells, followed by incubation for 2 h in the presence of 5 mM CaCl_2_. The wells were then washed with TBS containing 1 mM CaCl_2_ three times and blocked with 2% skim milk for 1 h. Mouse anti-recombinant human ANV polyclonal antibodies, which were prepared in our laboratory, were added to the wells, followed by incubation for 1 h. The wells were then washed and blocked with 2% skim milk for 15 min. HRP-labeled goat anti-mouse IgG (Dingguo, Beijing, China) was added to the wells, and, then, the plate was incubated for 60 min. The wells were then washed and developed by the addition of 140 L 0.03% (*w*/*v*) tetramethylbenzidine and 0.018% (*v*/*v*) H_2_O_2_ in citrate–acetate buffer (pH 5.0). Color development was stopped by the addition of 40 L of 2 N H_2_SO_4_. The absorbance of each well was then read at 450 nm.

Phospholipid binding dose analysis was carried out as follows. The amount and concentration of LbtA5 in each well were 50 μL and 50 μg/mL, respectively. The doses of PS were set to 0 μg, 0.3 μg, 0.6 μg, 0.9 μg, 1.2 μg, 1.5 μg, 1.8 μg, 2.1 μg, 2.4 μg, 2.7 μg, and 3 μg, and the other steps were the same as above.

The effect of the calcium ion concentration on LbtA5 binding phospholipids was determined as follows. The amount and concentration of LbtA5 in each well were 50 μL and 50 μg/mL, respectively. The PS dose was set to 2.5 μg, and the Ca^2+^ concentrations in the binding buffer were set to 0 mM, 0.3 mM, 0.6 mM, 0.9 mM, 1.2 mM, 1.5 mM, 2 mM, 2.5 mM, 3 mM, 4 mM, and 5 mM. The other steps were the same as above.

### 4.3. Cell Adhesion

Forty microliters of Col-IV (10 μg/mL) or LN (50 μg/mL) (Sigma-Aldrich, Shanghai, China) was added to each well to coat the 96-well plate and placed at 4 °C for 12 h. The remaining liquid in the 96- well plate was recovered, and 100 μL of 1% BSA was added to each well for 1 h. The plate was washed twice with PBS. PC12 and HT29 (Dingguo) cells were grown in RPMI-1640 or McCoy’s 5A (containing 10% FBS and 1% penicillin/streptomycin). All cells were maintained under sterile conditions at 37 °C and 5% CO_2_. PC12 cells and HT29 cells were co-incubated with LbtA5 and ANV in EP tubes, respectively. The protein concentration gradients were set at 0.01 nM, 0.1 nM, 1 nM, and 10 nM with 5 × 10^4^ cells per well. After incubation for 1 h, the cells were transferred to the above 96-well plate and incubated for 1 h at 37 °C. All the liquid was aspirated from the wells, and the wells were washed twice with PBS to remove unadhered cells. Then, 60 μL of 4% paraformaldehyde was added to each well, fixed for 10 min, and washed 2 times with PBS. Then, 60 μL of 0.1% crystal violet was added to each well, stained for 10 min, and washed 4 times with PBS. Then, 100 μL of 1% SDS was added to each well, the cells were lysed fully, and the absorbance value at 600 nm was measured with an enzyme marker. No Col-IV or LN was used as a control.

### 4.4. B16F10 Cell Culture and Cell Proliferation Assay

The murine melanoma cell line B16F10 was purchased from DingGuo. Cells were stored and recovered from cryopreservation in liquid nitrogen and cultured in DMEM plus 10% FBS, 50 mg/mL streptomycin, and 50 U/mL penicillin in a 5%-CO_2_ humidified atmosphere. B16F10 cells were incubated with MTT (Genview, Houston, TX, USA) for 4 h, and 100 μL of DMSO (Sigma-Aldrich) was added to each well for 0.5 h at 37 °C. The absorbance was evaluated at 570 nm using a microplate reader (BMG, Ortenberg Germany).

### 4.5. Wound Healing Assay

Cells were seeded in 6-well plates. A scratch wound was created in a confluent cell culture using a sterile 200 μL pipette tip. The scratched cells were washed twice with PBS and maintained in serum-free medium. Photographs were taken using an inverted microscope (Zeiss, Jena, Germany) at 0 and 24 h. The black lines indicate the borders of the scratches, and the distances were measured using ImageJ software.

### 4.6. Apoptosis Detection by Flow Cytometry

B16F10 cells were cultured as previously described. Briefly, the cells (2×10^5^ cells/well) were seeded in a 6-well plate for 24 h to allow cell adhesion. The protein concentrations of the experimental groups were set to 0.1 nM, 1 nM and 10 nM. The seeded cells were either treated with DMEM as a negative control or paclitaxel (10 μM) as a positive control. The cells were cultured for 24 h, and cell suspensions were prepared. All the liquid in the wells was collected and placed in a centrifuge tube together with the digested cells, centrifuged at 300× *g* for 5 min, and washed twice with PBS. Then, 500 μL of binding buffer was added, and the sample was resuspended and filtered with a 200-mesh nylon membrane. After 30 min of staining, the cells were analyzed by flow cytometry. All of the abovementioned experiments were repeated three times.

### 4.7. Western Blot

The cells were collected, and the total intracellular protein was extracted. Next, 80 μL of lysis solution (Beyotime) was added, and the sample was vortexed for sufficient lysis. The sample was centrifuged for 5 min (4 °C, 12,000 rpm), the supernatant was carefully aspirated and collected in another EP tube, and the total protein concentration was determined using the BCA method. After sample preparation, equal amounts of a sample protein (10 μg) were loaded onto an SDS–PAGE gel (10% separating gel, 5% stacking gel). After electrophoresis, the samples were transferred onto a nitrocellulose membrane. The membranes were blocked in blocking buffer for 1 h. After that, the membranes were incubated with primary antibodies against VEGF-A (Boster, Wuhan, China) and α-tubulin (Beyotime), followed by secondary antibodies coupled with horseradish peroxidase for 1 h. Reactive bands were detected with an ECL Western Blot Detection Kit (Boster) and visualized with a Tanon Imager program (Tanon, Shanghai, China). Non-saturated bands were selected to perform densitometric quantification using ImageJ software.

### 4.8. Mouse Model

Seven-week-old female C57BL/6J mice were obtained from Vital River (Beijing, China). All animal experiments were approved by the Animal Care and Use Committee of the College of Life Sciences, Beijing Normal University. Mice were injected with 100 μL of PBS containing 4 × 10^5^ B16F10 cells and injected into the right hindquarters through hypodermic injection. Mice were randomly divided into 6 groups (*n* = 5/group), and the first treatments were initiated on the 12th day after inoculation. The dosage units of drugs in animal experiments are usually mg/kg, and two concentration gradients (5 mg/kg and 10 mg/kg) were set up for the experiments. However, because the relative molecular masses of the two drugs are different, the molar amounts of ANV and LbtA5 were unified to ensure that the experimental results were not affected by the number of drug molecules. ANV (139 μmol/kg and 278 μmol/kg) and LbtA5 (139 μmol/kg and 278 μmol/kg) were injected intraperitoneally for 13 consecutive days, and 100 μL of PBS was injected as a negative control and DTIC (40 mg/kg DTIC) (Aladdin, Shanghai, China) as a positive control. Tumor volume was measured every day after the initial injection with calipers and determined as mm^3^ using the equation A × B^2^ × 0.52 [38], where A is the length (mm) and B is the width (mm). After 13 days, the mice were sacrificed, and melanoma tissues were processed for paraffin embedding and stained with hematoxylin and eosin (H&E) (Dingguo).

### 4.9. Immunohistochemistry (IHC)

Tissue blocks were cut into 5 μm slices, spread, and placed in a 60 °C incubator for 1 h to dry. The sections were dewaxed and rehydrated and washed with distilled water for 30 s. Antigen repair and endogenous peroxidase blocking were performed dropwise for 10 min, and the sections were washed with PBS. The sections were permeabilized with 1% Triton X-100 (Beyotime), blocked with 4% BSAT at room temperature for 30 min, and incubated overnight at 4 °C protected from light with primary antibody (Desmin). Then, the sections were incubated with the labeled secondary antibody (Alexa Fluor 555) at room temperature for 2 h. The sections were sealed with a blocker containing DAPI (Beyotime) and observed by inverted fluorescence microscopy at 340 nm for DAPI excitation and 555 nm for Alexa Fluor 555 excitation.

### 4.10. Preparation of FITC-Labeled Proteins and Immunofluorescence Assay

FITC (Sigma-Aldrich) master mix was prepared at a concentration of 1 mg/mL under low light. And 0.01 mg of fluorescein per milligram of ANV and LbtA5 protein was added to prepare the corresponding concentration of FITC working solution. Equal volumes of FITC and ANV or LbtA5 (1 mL each) were mixed one at a time and mixed well. After 36 h of binding, the labeled proteins were centrifuged at 10,000 rpm for 20 min, and a small amount of precipitate was removed and packed into MD34 dialysis bags (TaKara, Japan) and then fixed into beakers. Depending on the protein molecular weight, the labeled FITC-ANV or FITC-LbtA5 was eluted first, and the elution peaks were collected by UV meter readings.

The mouse melanoma model was constructed as described previously, and the mice were randomly divided into a negative control group (PBS), FITC-LbtA5 group (139 μmol/kg, 417 μmol/kg, 834 μmol/kg) and FITC-ANV group (139 μmol/kg, 417 μmol/kg, 834 μmol/kg). The mice were injected with protein solution in the tail vein and sacrificed after 30 min by decapitation. The mice were then sectioned, dewaxed, rehydrated, and sealed directly with DAPI blocker, and photographed for fluorescence observation (Zeiss).

### 4.11. Statistical Analysis

All numeric data are expressed as the mean ± SEM unless otherwise indicated. Experimental data were obtained from at least 3 independent replicate experiments. The significance between two groups was analyzed by two-tailed Student’s *t* test. Statistical analysis was performed using GraphPad Prism 8.0. *p* values of less than 0.05 were considered significant. * *p* < 0.05, ** *p* < 0.01, *** *p* < 0.001 and **** *p* < 0.0001.

## 5. Conclusions

Summarizing all experiments, we suggest that LbtA5 holds promise as a recombinant antimelanoma drug that can be more effective than ANV. In particular, we found that LbtA5 accumulated more in melanoma tissue than ANV, and the antiangiogenic effect was more pronounced, thus exerting an antitumor effect. In addition, ANV treatment can be combined with disintegrin LBT to enhance therapeutic efficacy.

## 6. Patents

The work reported in this manuscript has been declared as a China Invention Patent (registration number: 2023100584030).

## Figures and Tables

**Figure 1 ijms-24-03859-f001:**
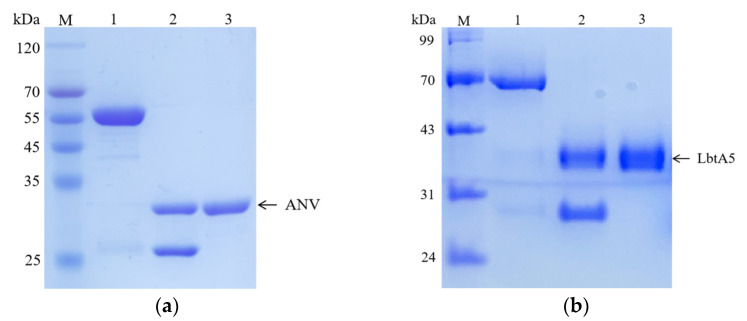
Analyses of the purified LbtA5 and ANV by SDS–PAGE on a 12% resolving gel. The gel was stained with Coomassie blue R-250. (**a**) Lane 1: purified GST-ANV (62.76 kDa); Lane 2: ANV (36.35 kDa) and GST isolated after PSP enzymatic digestion; Lane 3: purified ANV; (**b**) Lane 1: purified GST-LbtA5 (67.37 kDa); Lane 2: LbtA5 (40.96 kDa) and GST (26.43 kDa) isolated after PSP enzymatic digestion; Lane 3: purified LbtA5.

**Figure 2 ijms-24-03859-f002:**
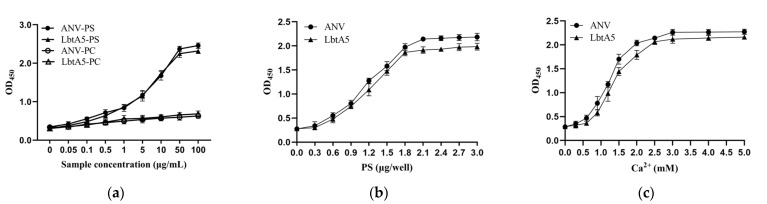
Phospholipid binding analysis of LbtA5 and ANV. (**a**) Binding analysis of ANV and LbtA5 to different cell membrane phospholipids; (**b**) Binding analysis of ANV and LbtA5 to different doses of PS; (**c**) Effect of different Ca^2+^ concentrations on the binding of ANV and LbtA5 to phospholipids.

**Figure 3 ijms-24-03859-f003:**
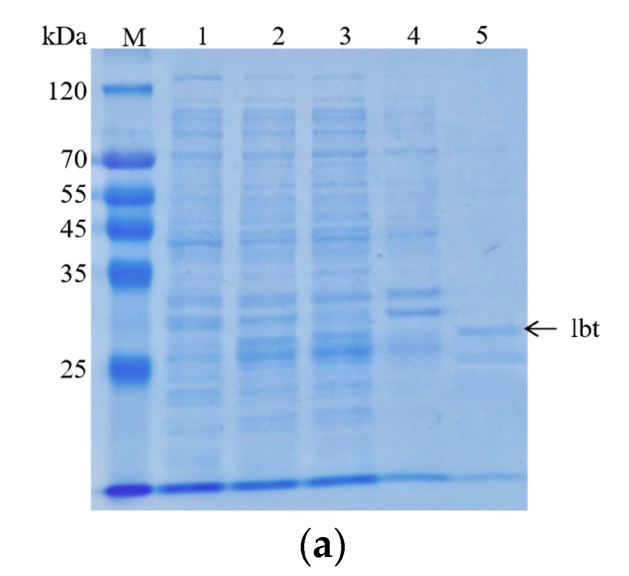
Analysis of PC12 and HT29 cell adhesion. (**a**) Analyses of the purified lbt by SDS–PAGE on a 12% resolving gel. The gel was stained with Coomassie blue R-250. Lane 1: pGEX-6p-1-lbt without IPTG induction; Lane 2: whole protein after sonication of the bacteriophage; Lane 3: supernatant protein after centrifugation. Lane 4: precipitated protein after centrifugation; Lane 5: 10 mM GSH-eluted lbt; (**b**,**c**) PC12 and HT29 cells were observed with an ordinary light microscope at 100×; (**d**,**e**) PC12 and HT29 cells treated with different concentrations of lbt, LbtA5, and ANV. After treatment, the cells were incubated at 37 °C and 5% CO_2_ for 24 h and photographed at 100× with an ordinary light microscope. Mean ± SD, *n* = 6, * *p* < 0.05, *** *p* < 0.001 using one-way ANOVA, both compared to control.

**Figure 4 ijms-24-03859-f004:**
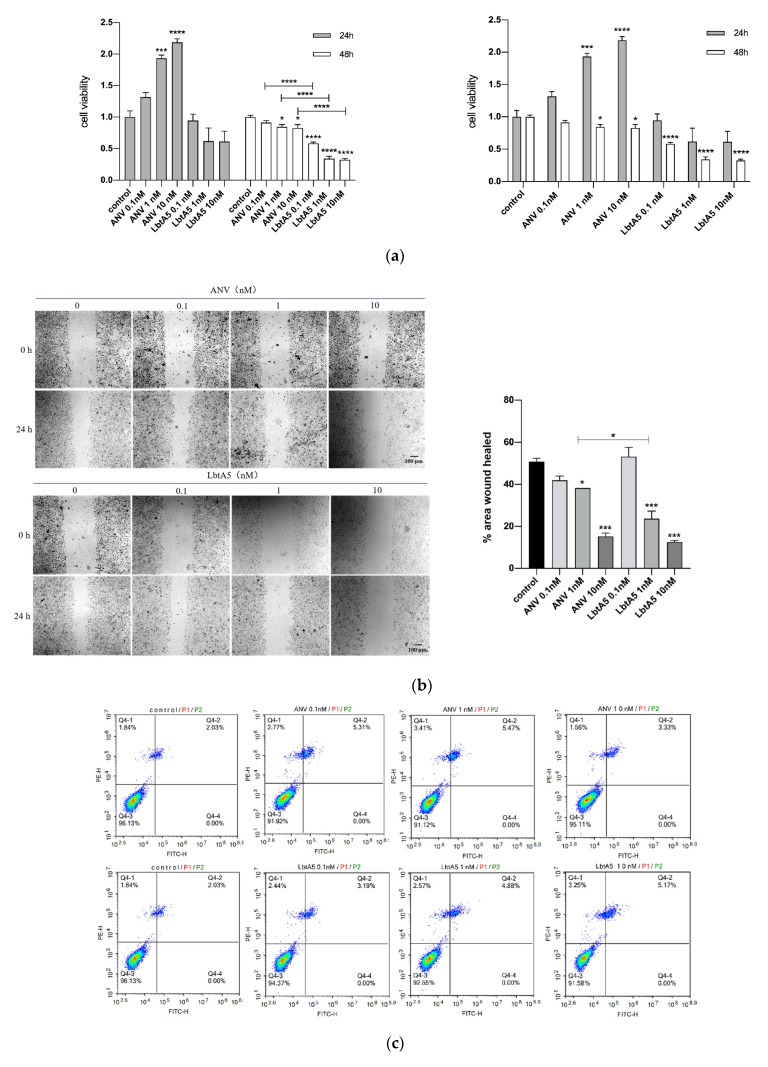
LbtA5 inhibits the proliferation and migration of melanoma B16F10 cells. (**a**) Effect of ANV and LbtA5 on the viability of B16F10 cells by MTT assays. One-way ANOVA, mean ± SEM, *n* = 6, * *p* < 0.05, *** *p* < 0.001, **** *p* < 0.0001 compared with the control or between groups. Cells were incubated at 37 °C and 5% CO_2_ for 24 h after different treatments, and, then, the absorbance values were measured at 570 nm by the MTT method. (**b**) Cell scratch assay to detect the effect of ANV and LbtA5 on the migration of B16F10 cells. ImageJ analysis of the 24-h change in scratch area. One-way ANOVA, mean ± SD, *n* = 3, *** *p* < 0.001, compared with 0 h or between groups; (**c**) Detection of the effect of ANV and LbtA5 on apoptosis of B16F10 cells by flow cytometry.

**Figure 5 ijms-24-03859-f005:**
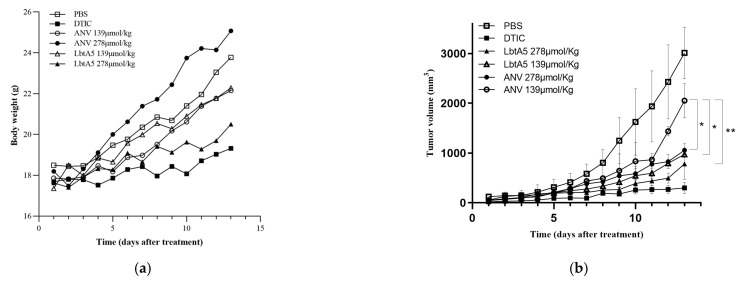
The antitumor effects of ANV and LbtA5 in mice bearing B16F10 melanomas. (**a**) Body weights of mice bearing B16F10 melanomas. Statistical analysis using the mean ± SD, *n* = 5; (**b**) Tumor volume comparison among different groups, * *p* < 0.05, ** *p* < 0.01; (**c**) Comparative histological analysis of tumor necrosis between different groups by H&E staining. After 13 days of continuous treatment, the mice were sacrificed, the tumor tissues were removed and prepared for sectioning, and H&E staining was performed and observed and photographed with an inverted microscope. The tumor tissues of the mice in the PBS group were the negative control group, and the DTIC group was the positive control group. The left side of each group was photographed at 100×, and the right side was photographed at 400× magnification of a local area in the same field.

**Figure 6 ijms-24-03859-f006:**
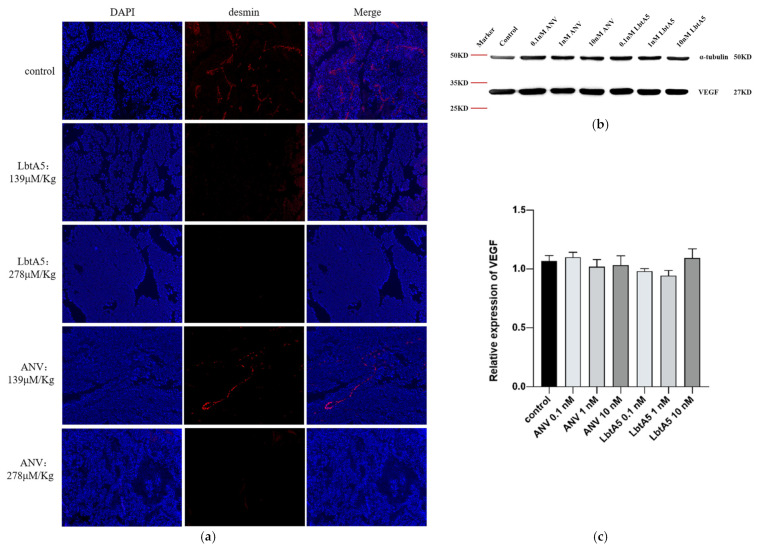
ANV and LbtA5 inhibited tumor angiogenesis. (**a**) Fluorescence images of tumor blood vessels. Vesicular staining (immunofluorescence) was performed and showed colocalization of desmin (red) in PBS-treated samples. The fluorescence distribution was photographed at a 555 nm wavelength with an inverted fluorescence microscope at 100×; (**b**) The effect of ANV and LbtA5 on the VEGF protein expression of B16F10 cells monitored by Western blot analysis; (**c**) Western blot grayscale data analysis (normalized using ImageJ software), VEGF relative expression = VEGF expression/internal reference protein expression, using one-way ANOVA, mean ± SEM, *n* = 3, ns for *p* > 0.05, all compared to control protein-free DMEM.

**Figure 7 ijms-24-03859-f007:**
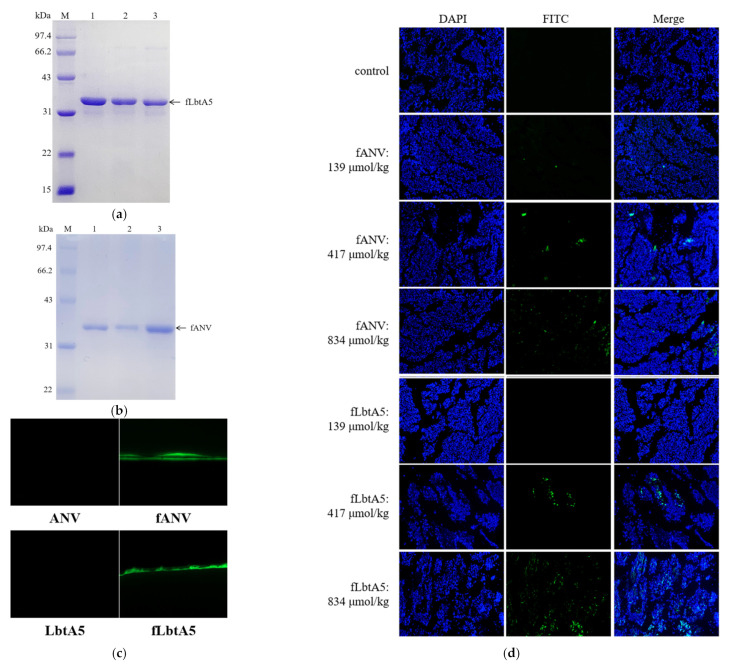
ANV and LbtA5 targeted tumor vessels. (**a**) Purified FITC-LbtA5 (40.68 kDa); (**b**) Purified FITC-ANV (36.28 kDa); (**c**) Fluorescence observation of FITC-labeled target proteins. FITC excitation wavelength: 488 nm; (**d**) FITC-ANV and FITC-LbtA5 were injected intravenously into mice bearing B16F10 solid tumors. Mice were sacrificed 30 min after tail vein injection, and tumor tissues were removed for section preparation. The control is the tumor tissue of the PBS group mice, which was excited at a 488 nm wavelength, and the fluorescence distribution was observed by inverted fluorescence microscope at 100×.

## Data Availability

The data presented in this study are available on request from the corresponding author.

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
