# Peer review of "Integrin Targeting Enhances the Antimelanoma Effect of Annexin V in Mice"

_ijms, 2023, doi:10.3390/ijms24043859_

Round 1

Reviewer 1 Report

The authors used recombinant DNA technology to synthetize Lebestatin-Annexin V (designated as LbtA5) fusion protein, which was successfully prepared with good stability and high purity, while retaining the dual biological activity of ANV and lbt. A series of various in vitro and in vivo tests showed that LbtA5 accumulates more in melanoma tissue than ANV, and that the anti-angiogenic effect is more pronounced, leading to the anti-tumor effect of LbtA5. In addition, ANV treatment can be combined with desintegrin LBT to enhance therapeutic efficacy.

The article is well-written, the experimental design is adequate, the results are clearly presented, the conclusions are properly made from the gained results and current knowledge regarding the investigated subject is supported with appropriate references.

Therefore, I recommend accepting this article for publication in IJMS after minor revision.

Minor points:

1.    Please mark on the gel-images (Figures 1, 3 and 7) the positions and names of the proteins of interest.

2.    The sentence “Mix well according to the enzymatic ratio of PSP to the target protein as previously explored in the lab” should not be in the results section (line 101).

3.    Certain parts of the Materials and methods (in fact, almost all subsections) are written in the present tense (e.g. transfer, incubate, aspirate, add, resuspend ...). Please write them in the form that is standard for writing a scientific paper.

4. line 513 – 800r – centrifugal force must be expressed in g

5.    Several times a capital or lowercase letter was mistakenly used when it shouldn't be (e.g. Malignant melanoma, Phosphatidylserine, c-terminus, image J…). The article also requires some improvement of the English language.

Author Response

Reponse to reviwer:

We thank the reviewers for their careful reading and approval of the article. Your comments were very valuable and helpful. We have carefully revised the manuscript in accordance with your comments.

  1.    Please mark on the gel-images (Figures 1, 3 and 7) the positions and names of the proteins of interest.

Response: Thanks for the suggestion, we have marked the relevant proteins with their position and name in the gel images.

  1.    The sentence “Mix well according to the enzymatic ratio of PSP to the target protein as previously explored in the lab” should not be in the results section (line 101).

Response: We have removed this sentence.

  1.    Certain parts of the Materials and methods (in fact, almost all subsections) are written in the present tense (e.g. transfer, incubate, aspirate, add, resuspend ...). Please write them in the form that is standard for writing a scientific paper.

Response: We thank the reviewer for the valuable comment. We have double-checked the manuscript and corrected the use of inappropriate tenses in sentences. This manuscript has been linguistically corrected in English by AJE(American Journal Expert) Language Editing Services, and the language Editing Certificate is attached at the end of the document.

  1. line 513 – 800r – centrifugal force must be expressed in g

Response: Thanks to the suggestion, we have changed the description of the centrifugation speed for collecting cells to 300g.

  1. Several times a capital or lowercase letter was mistakenly used when it shouldn't be (e.g. Malignant melanoma, Phosphatidylserine, c-terminus, image J…). The article also requires some improvement of the English language.

Response: We were really sorry for our careless mistakes. In our resubmitted manuscript, the typo is revised. And we do invited a friend of us who is a native English speaker to help polish our manuscript. See the revised version for more details. Thanks for your correction.

Reviewer 2 Report

The manuscript was aimed to show the anti-cancer effect of Lebestatin-Annexin V (LbtA5) fusion protein both in vitro and in vivo.  

Despite the accurate preparation of the fusion proteins and  appropriate characterization of their binding properties, the manuscript requires additional experiments to make this data publishable.   

1) The data and conclusions were based on the results obtained from single cancer cell line (B16F10 cells). I suggest the authors to expand their study by using a couple of melanoma cell lines and also recommend to use the non-transformed cells (as a negative control) for at least in vitro assays. Indeed, the expression of Annexin V in this particular cell line in unknown (was unshown) and non-transformed cells included in the study will be very informative for the in vitro assays. 

2)  Anti-cancer agent was included for the xenograft experiments only. However, all in vitro assays shown in Figure 4, lack the positive control.

3) The number of apoptotic (Annexin V-positive) cells shown by FACs analysis is very low (as shown in Figure 4C). However, this might be due to the unappropriate design of experiments. Indeed, the probability is very high that LbtA5 might compete with Annexin V during Annexin V/PI staining. To exclude this possibility, the authors have to run additional experiments and also include the positive control for this particular set of experiments.

4) Title of 2.4. requires editing since the data shown in Figure 5 illustrates necrotic changes both in LbtA5 and ANV-treated animals.  Moreover, both LbtA5 and ANV-treated tumors exhibited very similar histomorphological changes, including tumor necrosis, as shown in Figure 5. Moreover, the data presented in Figure 5 lacks quantification. Despite this fact, the authors argue about the most pronounced effect of LBTA5 on tumor tissue when compared with ANV.   

Author Response

Response to reviewer:

1) The data and conclusions were based on the results obtained from single cancer cell line (B16F10 cells). I suggest the authors to expand their study by using a couple of melanoma cell lines and also recommend to use the non-transformed cells (as a negative control) for at least in vitro assays. Indeed, the expression of Annexin V in this particular cell line in unknown (was unshown) and non-transformed cells included in the study will be very informative for the in vitro assays.

Response: 

Thank you for your comments. It is indeed interesting to explore the biological effects of LbtA5 fusion protein on other types of melanoma cell lines, as stated by the reviewers. We believe that these experimental results will bring us novel findings and provide new scientific insights. However, due to the limited time to revise the paper, the purification of AMV and LbtA5 proteins will take at least 2 days, and no other melanoma cell lines are currently available in the laboratory. Therefore, it is difficult to carry out this work in the short term, and we apologize for this.

Currently, there is more than one known melanoma cell line, and in addition to B16F10, there are cell lines such as A375. Although they are all melanoma cell lines, each has its own characteristics and is different from the other. Therefore, can we say that the biological effect of LbtA5 fusion protein on other melanoma cell lines should be different from that of B16F10, even in terms of the mechanism of action, which needs to be further investigated.

This manuscript focuses on the biological effects and mechanisms of B16F10 cell lines and their derived xenogeneic mouse tumor models. Therefore, can we change the title of the manuscript to “Integrin targeting enhances the anti-B16F10 melanoma effect of Annexin V in mice”? We are grateful to the reviewer for the suggestions and we will carry out the studies suggested by the reviewer in our future work and will publish them.

2)  Anti-cancer agent was included for the xenograft experiments only. However, all in vitro assays shown in Figure 4, lack the positive control.

Response:

Thank you for your comment. For cellular level assays it is more difficult to determine which substance is most suitable as a positive control when the mechanism of action is not clear. The purpose of MTT and cell migration is to compare the effects of ANV and LbtA5 on B16F10 cells, and we have set up a negative control. The results of MTT experiments illustrated that LbtA5 had a greater effect on B16F10 cell viability than ANV after 48 hours. The results of cell scratching illustrated that 1 nM of LbtA5 was more effective than ANV in inhibiting cell migration and was significantly different. In addition, as shown in the published articles below, these cellular level experiments allow for no positive control to be set, and a negative control would reflect the effect of the substance to be tested on specific tumor cells.

(1)Kuo CH, Wu YF, Chang BI, Hsu CK, Lai CH, Wu HL. Interference in melanoma CD248 function reduces vascular mimicry and metastasis. J Biomed Sci. 2022;29(1):98. Published 2022 Nov 18. doi:10.1186/s12929-022-00882-3 (Page5, Figure 1a, IF=12.771, JCR Q1)

(2)Li JK, Zhu PL, Wang Y, et al. Gracillin exerts anti-melanoma effects in vitro and in vivo: role of DNA damage, apoptosis and autophagy. Phytomedicine. 2023;108:154526. doi:10.1016/j.phymed.2022.154526 (Page3, Figure 1c, IF=6.656, JCR Q1) 

3) The number of apoptotic (Annexin V-positive) cells shown by FACs analysis is very low (as shown in Figure 4C). However, this might be due to the unappropriate design of experiments. Indeed, the probability is very high that LbtA5 might compete with Annexin V during Annexin V/PI staining. To exclude this possibility, the authors have to run additional experiments and also include the positive control for this particular set of experiments.

Response:

Thank you for your valuable comments. Our procedure for performing apoptosis assay experiments is as follows: (1) The target protein (LbtA5) was added to the cell culture system and co-incubated for 24 hours; (2) The culture medium was discarded and washed once with PBS; (3) Add trypsin to digest the cells, collect all the liquid in the wells, centrifuge at 300g for 5 min, and wash twice with PBS, resuspend with binding buffer and filter; (4) After sufficient washing, Annexin V dye or PI dye was added. Theoretically, there is no residual LbtA5 in the experimental system before adding the dye. Therefore, we believe that there is no problem of competitive binding of LbtA5 to Annexin V.

Cell morphology is significantly changed during apoptosis. Therefore, we observed whether ANV and LbtA5 had any effect on B16F10 cell morphology. In the experimental groups, B16F10 cells were treated with different concentrations of ANV and LbtA5, and the negative control was not treated with protein. The morphology of B16F10 cells at 0h and 24h was observed by inverted microscopy. The experimental results are shown in the figure below. Both control and experimental cells could grow against the wall with clear outline and good refractive property. LbtA5 in a certain concentration range had no significant effect on the morphology of B16F10 cells. In summary, based on cell morphology observations and FACs analysis, it was shown that neither ANV nor LBTA5 caused apoptosis in B16F10 cells.

In addition there are now also experimental results from published articles showing that ANV does not cause apoptosis in B16F10 cells.

Figure. Morphological effects of exogenous addition of ANV and LbtA5 on B16F10 cells. The control was protein-free DMEM. B16F10 cells were incubated at 37℃, 5 %CO2 for 24 h after different treatments and photographed with an inverted microscope at 100×.

4) Title of 2.4. requires editing since the data shown in Figure 5 illustrates necrotic changes both in LbtA5 and ANV-treated animals.  Moreover, both LbtA5 and ANV-treated tumors exhibited very similar histomorphological changes, including tumor necrosis, as shown in Figure 5. Moreover, the data presented in Figure 5 lacks quantification. Despite this fact, the authors argue about the most pronounced effect of LBTA5 on tumor tissue when compared with ANV.   

Response:

Thanks to the reviewers' comments, we quantified some of the data in Figure 5 to  reflect the significance of tumor volume. The quantification results showed that LbtA5(139 μmol/Kg) was significantly more effective than ANV(139 μmol/Kg) for treatment. A large number of images of tissue sections were taken in our work, and we only put in some of them due to the limitation of article length. The HE staining results we observed illustrated that LbtA5 caused more tumor necrosis than ANV. Thus, the HE staining and tumor volume data suggest that LbtA5 has a stronger anti-melanoma effect than ANV. Figure5 corresponds to the result sections 2.5 and 2.6, not 2.4.

The differences between LbtA5 and ANV-treated melanoma are mainly reflected in two aspects: (1) Immunofluorescence results showed that LbtA5 more inhibited tumor tissue angiogenesis. (2) The same concentration of LbtA5 accumulated more in tumor tissues than ANV. According to published articles, desintegrin lbt is an inhibitor of angiogenesis, which is one of the essential aspects of tumor growth. Our results illustrate the stronger anti-melanoma effect of ANV coupled to desintegrin lbt. This may be an effective strategy for the treatment of metastatic melanoma. In the future, we will continue to investigate the specific mechanisms of LbtA5 and ANV anti-melanoma in depth, explore the relationship between lbt, LbtA5 and ANV and tumor angiogenesis, and publish the obtained data as an article.

Round 2

Reviewer 2 Report

The authors responded properly to the comments and suggestions. The quality of the manuscript was improved. Some experiments are still missing due to the limited time-frame for the revision. Therefore, the authors suggested to rename the manuscript to the "Integrin targeting enhances the anti-B16F10 melanoma effect of Annexin V in mice" and I support this suggestion.